# The Complete Chloroplast Genome of Two Important Annual Clover Species, *Trifolium alexandrinum* and *T. resupinatum*: Genome Structure, Comparative Analyses and Phylogenetic Relationships with Relatives in Leguminosae

**DOI:** 10.3390/plants9040478

**Published:** 2020-04-09

**Authors:** Yanli Xiong, Yi Xiong, Jun He, Qingqing Yu, Junming Zhao, Xiong Lei, Zhixiao Dong, Jian Yang, Yan Peng, Xinquan Zhang, Xiao Ma

**Affiliations:** 1College of Animal science and Technology, Sichuan Agricultural University, Chengdu 611130, China; yanlimaster@126.com (Y.X.); xiongyi95@126.com (Y.X.); yuqinggzu93@126.com (Q.Y.); junmingzhao163@163.com (J.Z.); lxforage@126.com (X.L.); dongzhixiao94@126.com (Z.D.); iwanwin@126.com (J.Y.); pengyanlee@163.com (Y.P.); 2State Key Laboratory of Exploration and Utilization of Crop Gene Resources in 10 Southwest China, Key Laboratory of Biology and Genetic Improvement of Maize in 11 Southwest Region, Ministry of Agriculture, Maize Research Institute of Sichuan 12 Agricultural University, Chengdu 600031, China; hejunyer@outlook.com

**Keywords:** chloroplast genome, *Trifolium*, divergence time, IR lacking, rearrangement, repetitive events

## Abstract

*Trifolium* L., which belongs to the IR lacking clade (IRLC), is one of the largest genera in the Leguminosae and contains several economically important fodder species. Here, we present whole chloroplast (cp) genome sequencing and annotation of two important annual grasses, *Trifolium alexandrinum* (Egyptian clover) and *T. resupinatum* (Persian clover). Abundant single nucleotide polymorphisms (SNPs) and insertions/deletions (In/Dels) were discovered between those two species. Global alignment of *T. alexandrinum* and *T. resupinatum* to a further thirteen *Trifolium* species revealed a large amount of rearrangement and repetitive events in these fifteen species. As hypothetical cp open reading frame (ORF) and RNA polymerase subunits, *ycf1* and *rpoC2* in the cp genomes both contain vast repetitive sequences and observed high Pi values (0.7008, 0.3982) between *T. alexandrinum* and *T. resupinatum*. Thus they could be considered as the candidate genes for phylogenetic analysis of *Trifolium* species. In addition, the divergence time of those IR lacking *Trifolium* species ranged from 84.8505 Mya to 4.7720 Mya. This study will provide insight into the evolution of *Trifolium* species.

## 1. Introduction

*Trifolium* L. (Leguminosae, Fabaceae), one of the largest genera in the Leguminosae, contains several important fodder species, such as *T. repens* (white clover), *T. pratense* (red clover), *T. alexandrinum* (Egyptian clover), *T. resupinatum* (Persian clover) amongst others [1]. *Trifolium* species are also widely grown as green manure crops, and about 11 species, including *T. alexandrinum* and *T. resupinatum*, were introduced to the subtropical zone of east Asia and have been reported to be excellently adapted to saline-alkali soil thus useful for agricultural production [2]. *T. alexandrinum* is generally grown as an annual winter legume fodder crop in the Middle East, Mediterranean and the Indian subcontinent. Its aerial part can be used for cattle feed and the seeds are used as an antidiabetic treatment [3]. Furthermore, *T. alexandrinum* also contributes to soil fertility and improves soil physical characteristics [3]. *T. resupinatum*, an annual, prostrate or semi-erect branched legume, can supply highly palatable and nutritive pasture and hay [4]. What’s more, it also has economic significance for the ornamental and landscape industries [4].

As an important part of plant organelles and photosynthetic organ, chloroplast (cp) has played an irreplaceable role in plants [5]. The cp genomes are not only essential for the study of plants light system for potentially improving the photosynthetic capacity and thus increasing plant yield, but are also commonly employed for phylogenic study for their maternal inheritance and highly conserved genomic structure [6]. The cp genome has a typically covalently closed circular molecule structure including a small single-copy region (SSC), a large single-copy region (LSC) and two almost identical inverse repeats (IRs) regions [5]. A typical cp genome contains approximately 130 genes. Many of them participate in photosynthesis, some others also encode proteins or function in regulating gene transcription [6].

Possessing the function of replication initiation, genome stabilization and gene conservation [7], the IR regions with the average length between 10 Kb to 76 Kb were found in almost families in the angiosperm plants and some gymnosperm and fern genus [8]. However, a lack of IRs was found in the clover genus (*Trifolium*) of the legume family (Leguminosae) and coniferous cp genomes, such as *Medicago* L., *Melilotus* Miller, *Ononis* L. and so on [2]. A previous study has shown that differential elimination of genes from the IR regions may result in IR lacking in Ulvophyceae [9]. Meanwhile, gene loss including *ycf2* and *psbA* lead to IR lacking from cp genomes of *Cedrus wilsoniana* and *C. deodara* [10]. However, what causes the IR lack of cp genomes in *Trifolium* species still remains to be elucidated.

The lack of IR generates a highly rearranged cp genome, thus experiencing gene losses and inversions in the SSC or LSC regions [11]. Furthermore, quadruple of repeated DNA compared to related legume species is found to typically exist in IR lacking subclover (*T. subterraneum*) based on previous studies [12]. However, this unusual repeat-rich cp genome structure was also reported in other IR lacking *Trifolium* species, such as *T. repens*, *T. meduseum*, *T. semipilosum* and so on [13]. Those repetitive structures might be related to the sequence rearrangement of cp genomes via intra homologous recombination [8] in some angiosperm lineages like the Campanulaceae [14]. However, what causes the repeat-rich cp genomes structures of the *Trifolium* species is still worth studying, although previous studies have suggested that it may be related to the nuclear genes [13].

Insights gained from cp genome sequences and structure have improved the revelation of variation among plant species and made significant contributions to phylogenetic analysis [6]. Leguminosae are accepted to have flourished since the Cretaceous period and the phylogenetic relationships among some *Trifolium* species were well estimated using 58 protein-coding genes in cp genomes [13]. However, two important annual *Trifolium* species, *T. alexandrinum* and *T. resupinatum* have not been included in any previous study. Variation among different species could provide a fascinating glimpse into the understanding of plant biology and diversity [6]. Here, cp genomes of *T. alexandrinum* and *T. resupinatum* were sequenced and annotated. We compared the sequence differences caused by nucleotide diversity (Pi), In/Del and repetitive sequences, as well as the evolution pressure reflected by non-synonymous/synonymous (Ka/Ks) between these two species. Furthermore, they were compared with further thirteen IR lacking congeneric species and divergent times were estimated. This study provides insights into the evolution of IR lacking cp genomes.

## 2. Results

### 2.1. Features of the T. alexandrium and T. resupinatum cp Genomes

More than 20 million ReadSum (pair-end reads) were yielded from *T. alexandrium* and *T. resupinatum*, with the Q20 and Q30 (the percentage of bases whose mass value is greater than or equal to 20, 30) higher than 94% and 87%, respectively. We assembled them successfully based on the alignment of paired-end sequences to the reference of *T. medeseum* (Figure 1). The cp genomes of *T. alexandrium* and *T. resupinatum* were detected with a lack of IR and have a size of 148,545 bp and 149,026 bp, respectively (Table 1). The GC content observed in the two cp genomes was 34.09% and 33.80% overall, and 37.05% and 36.64% in coding sequences (CDS). A total of 112 and 109 genes were consisted in the complete cp genomes of *T. alexandrinum* and *T. resupinatum*, which contains 31 and 37 tRNA, 75 and 66 mRNA, and 6 rRNA, and 13 and five genes possessing introns, respectively. In particular, there were two *rrn* 16 genes in each of the *Trifolium* species with the unidentical sequences.

There were 46 genes related to photosynthesis in cp genomes of *T. alexandrinum* and *T. resupinatum* (Table 2), of which four genes *psbN*, *atpF*, *ndhA* and *ndhB* were specific for *T. alexandrinum*. These genes include the ones encoding subunits of Rubisco, subunits of photosystem I, subunits of photosystem II, subunits of ATP synthase, cytochrome b/f complex, c-type cytochrome synthesis and subunits of NADH dehydrogenase. Thirty-one genes were related to self-replication, including four ribosomal RNA genes and 27 transfer RNA genes, in which *trnT-CGU* was unique in *T. alexandrinum*. Besides, ten genes encoded ribosomal proteins and twelve were associated with transcription. Among them, *rpl2* and *rpoC1* were unique in *T. alexandrinum*. Furthermore, three genes *clpP*, *accD* and *ycf3* with other functions were particular for *T. alexandrinum* (Appendix A).

Introns are generally not subject to natural selection thus theoretically accumulate more mutations than exons. In this study, a total of seven genes (*atpF*, *clpP*, *ndhA*, *ndhB*, *rpoC1*, *rps18* and *tRNA-CGU*) only contained an intron in *T. alexandrinum* (Appendix A). Other five genes *tRNA-UAA*, *tRNA-UAC*, *tRNA-UGC*, *tRNA-UUC* and *tRNA-UUU* all had an intron in *T. alexandrinum* and *T. resupinatum.* The exons length of those five genes was more conserved compared with the intron. In particular, *ycf3* had two introns in *T. alexandrinum*.

### 2.2. Repeat Sequences Analysis

Scattered repetitive sequences (palindrome repeats and direct repeats) and simple sequence repeats (SSRs) were analyzed respectively. A total of 1941 scattered repetitive sequences in the *T. alexandrinum* cp genome were annotated, which was greater than *T. resupinatum* (1250). The percentages of palindrome repeats (type P, 50.49%, Figure 2B) of *T. alexandrinum* were slightly larger than *T. resupinatum* (46.4%). A total of 370 (Figure 2A) and 383 SSRs (sizes ranged from 8–81 bp and 8–36 bp) were predicted in *T. alexandrinum* and *T. resupinatum* and 30.54% and 23.24% of them were distributed in genic regions. In particular, the majority of SSRs were located in *ycf1* (18 for *T. alexandrinum* and *T. resupinatum*), followed by *rpoC2* (11 for *T. resupinatum* and 9 for *T. alexandrinum*). Mononucleotide repeats were dominant (65.41% in *T. alexandrinum* and 74.93% in *T. resupinatum*), followed by trinucleotide repeats (25.68% in *T. alexandrinum* and 22.19% in *T. resupinatum*), in which the polyadenine repeats (poly A, 37.34% for *T. resupinatum* and *35.95%* for *T. alexandrinum*) and polythymine (poly T, 36.55% for *T. resupinatum* and 37.84% for *T. alexandrinum*) were much more than guanine (G) or cytosine (C) repeats (less than 1.35%). A total of 24 SSRs were identified to be shared by *T. alexandrinum* and *T. resupinatum* (Appendix A; Figure 2A). The common repeat sequences larger than 30 bp with the longest length of 117 bp was showed in Figure 2C.

### 2.3. Relative Synonymous Codon Usage Analysis

Relative synonymous codon usage analysis (RSCU), which is considered to be a combination result of natural selection, species mutation and genetic drift, was analyzed (Appendix A; Appendix A). The RSCU value for initiation codon AUG was 2.9745 in *T. alexandrinum* and 2.9721 in *T. resupinatum*. The values of three termination codons UAA, UAG and UGA were 1.6215, 0.5676 and 0.8109 in *T. alexandrinum*, and 1.5909, 0.5454 and 0.8637 in *T. resupinatum*. The codons with an RSCU value greater than one were considered to be a greater codon frequency. 46.97% (31 of 66, include three termination codons) of the codons were with the greater codon frequency both in *T. alexandrinum* and *T. resupinatum*, in whih 93.55% (29 of 31) prefers A or U in the third sites. In the other codons with RSCU values less than one (including one), C or G were more common in the third position (88.57%, 31 of 35).

### 2.4. Ka/Ks, Single Nucleotide Polymorphisms (SNPs) and Insertions/Deletions (In/Dels)

Single nucleotide polymorphisms (SNPs), mainly including transversion (Tv) and transition (Tn), along with insertions/deletions (In/Dels) could lead the non-synonymous (Ka) or synonymous (Ks) substitution. SNPs and In/Dels in every gene varied from 1 (*ndhE* and *psaC*) to 677 (*atpB*) with a total of 8560. Additionally, more In/Dels, Tn and Tv were detected in intergenic regions (5.66%, 17.11% and 38.70%) than genic regions (3.05%, 10.40% and 25.08%) (Figure 3; Appendix A). The 66 shared protein-coding genes with variations were used to calculate the Ka and Ks (Appendix A). The values of Ka and Ks ranged from 0 (*ndhE*, *petD*, *psaI*, *psbA*, *psbB* and so on) to 3.0151 (*rps4*) and 0 (*petG*, *petN*, *ndhD*, *psaJ*, *pabK*, *rpl23* and *rpl36*) to 2.9415 (*rps8*). Except for seven genes with Ks = 0, the 59 shared genes were used to calculate Ka/Ks, which varied from 0 (*ndhE*, *psbZ*, *psbA*, *psbJ* and so on) to 3.7723 (*rps4*, Figure 4), respectively. Seven genes including *rps4*, *rpoC2*, *ndhG*, *ccsA*, *ndhF*, *rpoA* and *psaC* have Ka/Ks values above one, implying positive selection on these genes. The Pi values calculated by 96 common genes of *T. alexandrinum* and *T. resupinatum* were from 0 to 0.7867 (*trnl-CAU*). Twenty-one genes had a Pi values of 0, among which nineteen were tRNA. What’s more, the nine genes with Ka/Ks above one also possessed relatively high Pi values (Figure 5; Appendix A).

### 2.5. Whole cp Genome Comparison with Other Trifolium Species

In order to examine the sequence divergence of *Trifolium* genus and further shed light on the evolutionary events, such as gene mutation, rearrangement and gene loss, cp genomes of fifteen IR lacking species (*T. grandiflorum*, *T. hybridum*, *T. lupinaster*, *T. occidentale*, *T. semipilosum*, *T. aureum*, *T. boissieri*, *T. glanduliflerum*, *T. strictum*, *T. repens*, *T. pratense*, *T. subterraneum*, *T. meduseum*, *T. alexandrinum* and *T. resupinatum*) were compared. The results showed that the size of cp genomes of these IR lacking species ranged from 121,178 bp (*T. pratense*) to 149,026 bp (*T. resupinatum*), with an average of 134,062 bp (Table 1). The GC content of those fifteen species changed from 33.80% to 37.20% in whole cp genome with the mean value of 34.99%, and 35.71% to 37.34% in CDS with the mean of 36.55%. Only minor variations were detected in the total numbers of genes, tRNA and mRNA among the selected species. *T. pratense* possessed the smallest numbers of tRNA (28), mRNA (58) and total number of genes (90). Furthermore, abundant gene rearrangements at the cp genome level were detected among fifteen *Trifolium* species using MAUVE program and the *T. resupinatum* as the reference sequence (Figure 6).

### 2.6. Phylogenetic Divergence Time Estimation

The 41 protein-coding genes shared in cp genomes of the 25 species (23 of Papilionoideae, one of Caesalpinioideae and one of Mimosaceae) were subjected to phylogeny analysis and divergence times estimation (Figure 7A). The topological structure of phylogenetic tree was almost consistent with the classification of Leguminosae with strong bootstrap support. Three subfamilies of Leguminosae, Papilionoideae, Caesalpinioideae and Mimosaceae were clearly separated. Furthermore, two genes *ycf1* and *rpoC2*, which both contain vast repetitive sequences and high Pi values (0.7008, 0.3982) between *T. alexandrinum* and *T. resupinatum*, were also used to construct the phylogenetic trees. The result showed that the topological structure of the phylogenetic relationship of *Trifolium* species based on *ycf1* (Figure 7B) and *rpoC2* (Figure 7C) was almost in accordance with the phylogenetic tree constructed using 41 shared genes. Each Section of *Trifolium* was clearly divided based on *ycf1* and *rpoC2* genes. The *Trifolium* species of “refractory clade” (including Section Lupinaster, Trifolium, Tricocephalum, Vesicastrum and Trifoliastrum) were grouped together in Figure 7C. In Figure 7B, however, Section Trifolium, Tricocephalum, Vesicastrum and Trifoliastrum were grouped into one clade, and another clade consisted of Section Lupinaster, Paramesus and Subg. Chronosemium. *Trifolium* species split from *Medicago* species during the Early Cretaceous (116.1575 Mya) and the divergence time of those fifteen IR lacking *Trifolium* species ranged from 84.8505 Mya to 4.7720 Mya (Figure 7).

## 3. Discussion

### 3.1. Genome Feature of T. alexandrinum and T. resupinatum and Comparison with Other IR Lacking Trifolium Species

In this study, we sequenced and annotated cp genomes of two IR lacking species *T. alexandrinum* and *T. resupinatum*, identified the repeats and hotspot genes within the cp genomes, and constructed the phylogenetic tree along with other cp genomes. Our results added information about cp genomes of *Trifolium* species and provided new insights into the evolution study of IR lacking species.

The two cp genomes sequenced in this study revealed 75 and 66 protein-coding genes in *T. alexandrinum* and *T. resupinatum*, respectively (Table 1). There were nine unique genes in the *T. alexandrinum* cp genome which were absent in *T. resupinatum*, which might have been transferred from the cp genome to the nucleus genome of *T. resupinatum* in the evolutionary process of the species. A similar result was revealed in tobacco with an absent *accD* gene, which was essential for leaf development [15,16]. The *rpl2* gene, which was absent in *T. resupinatum*, has been completely or partially transferred to the nucleus genome of some legumes like soybean and *Medicago* [17]. Besides, the loss of two *ndh* genes (*ndhA* and *ndhB*) was usually related to the nutritional status and rearrangement in most angiosperm species [18]. Although gene loss of *atpF* [19], *psbN* [19], *rpoC1* [19] and *ycf3* [20] was also found in other species, however, the detailed reason for that gene loss remained to be explained.

Compared to other IR containing cp genomes of angiosperms, the number of protein-coding genes of *Trifolium* species is less conserved [21,22]. It might be related to the fact that IR lacking will lead to an extensively arranged cp genome thus causing diverse genes loss [11]. According to Millen et al. [23], the vast majority of cp genomes of angiosperms held in shared 74 coding-protein genes but other five genes (*accD*, *ycf1*, *ycf2*, *rpl23* and *infA*) only existed in some specific species. *infA* gene, which codes for translation initiation factor 1 (IF1), was defunct in all the listed fifteen *Trifolium* species. Considered as the most transferable gene in cp genome, *infA* was in existence in about 24 angiosperm lineages including *Trifolium* species, and related *Medicago* species [24,25]. It is worth noting that there are two *rrn16* gene in each of *Trifolium* species sequenced in this study, and this phenomenon was also reported in cp genomes of *T. strictum* [13] and *T. glanduliferum* [13]. Furthermore, we also found that one of the *rrn16* and *rrn23* was partially overlapped. Indeed, no similar phenomenon was found in other *Trifolium* species. Previous studies have shown that *Cyanobacteria*, *Chlamydomonas reinhardtii*, *Cyanophora paradoxa*, *Zea mays*, *Oryza sativa* and *Arabidopsis thaliana* can start and stop transcription anywhere in the cp genome [26]. Chloroplasts are known to originate from ancient *Cyanobacteria*, this transcription property may be conserved in some species like *T. alexandrinum* and *T. resupinatum*, so that the rRNA genes of those two species may gain potential transcription ability. On the other hand, the cp genomes of most plant species are small and in order to avoid costs, genes overlapping happened. The *Ycf1* (*hypothetical chloroplast reading frame no. 1*) gene, generally with the premature stop codons in the CDS thus be defined as pseudogene in other angiosperm [27], has undergone processes of accelerated mutation rate, decreased GC content, and decreased secondary structure stability [28]. In this study, the *ycf1* gene detected in all the fourteen *Trifolium* species (except *T. hybridum*) contains a normal stop codon and is able to code protein. Two genes *ycf4* (*hypothetical chloroplast reading frame no. 4*) and *rps16* (*ribosomal protein S16*), which were found in most cp genomes of angiosperm and some relic plant like *Amborella trichopoda* [29], are not present in the two *Trifolium* species cp genomes sequenced in this study, and *rps16* is not present in all the fifteen *Trifolium* species. The *rps8* (*ribosomal protein S8*) gene was found without stop codons in *Medicago truncatula* and *M. sativa* [25], while it possesses a stop codon in the fifteen *Trifolium* cp genomes. In the gene *ndhB*, no internal stop codon was detected in the fourteen (except *T. resupinatum*) *Trifolium* cp genomes, which is inconsistent with many legume cp genomes whose *ndhB* gene contains an internal stop codon [25].

### 3.2. Relative Synonymous Codon Usage Analysis (RSCU)

The unequal using frequencies of synonymous codons detected in most sequenced genomes was termed synonymous codons usage bias [30], and is now considered crucial in shaping gene expression and cellular function [31]. RSCU indicates the relative probability of a particular codon encoding the synonymous codon of the corresponding amino acid. In this study, 93.5% of the codons were found prefer A/U in third position, similar to that of *M. sativa* [25] and *Gossypium* [32]. This phenomenon could be attributed to the fact that dicotyledon prefers to the A/U-end codons and manifests a potential force in molecular evolution of *Trifolium* species: mutation and natural selection.

### 3.3. SSRs and Large Repeat Sequences

Given a matrilineal inheritance feature, rich number of tags and low frequency of genetic recombination, cpSSRs (chloroplast simple sequence repeats) are considered an efficient molecular marker in genetic variety analyzing, population structure studying, species identification and phylogeny analysis [33]. The SSRs identified in *T. alexandrinum* and *T. resupinatum* were poly(A)/(T), which was consistent with the majority of plant family [34,35,36]. Although there have many studies reported on the application of cpSSRs in plant genetic diversity analysis, the important potential of cpSSRs in studying the ecological and evolutionary processes of wild materials of plants and their related species still need to be recognized [37]. Therefore, the cpSSRs of the two *Trifolium* species detected in this study can be used to evaluate genetic relationships among different species and to detect polymorphism of *Trifolium* species and their relatives at the population level.

Repetitive sequences related to the continuously self-replicating of genetic material in the process of evolution, thus indicating the greatly expanded and enriched genetic information [38]. The present study revealed a relatively high repetitive percentage (7.325%, Table 1) in the cp genomes of fifteen IR lacking species, which was higher than IR lacking *Tydemania expeditionis* (0.4%) and *Bryopsis plumose* (2.4%) [7]. The number of repeated sequences in the cp genome are associated with rearrangement in some species [14]. However, the driving force of the repetitive sequence was predominantly related to nuclear genes and genomic recombination [12], such as homologous recombination and microhomology-mediated break-induced replication acting on more than 50 bp and less than 30 bp repeats, respectively. Known as “hotspots” for variation [39], *ycf1* and *rpoC2* possessed high values of Pi (Figure 5) and the majority of repetitive sequences in *T. alexandrinum* and *T. resupinatum*. Therefore, these two genes could have suffered from selection pressure and could be used for phylogenetic analysis and population genetic study of *Trifolium* species.

### 3.4. Sequence Divergence and Hotspots

Point mutation was generally more common than frame shift for natural mutation [40]. As expected, more SNPs (21963, 6618 Tn and 15345 Tv; Appendix A) than In/Del (2097) were found between *T. alexandrinum* and *T. resupinatum*. What’s more, 60% of them occurred in intergenic regions, which was consistent with the hypothesis that CDS had a slower rate of evolution compared with CNS [41]. Furthermore, minor SNPs (159 between *Oryza sativa* and *O. nivara* [42], 330 between *Citrus sinensis* and *C. aurantiifolia* [43] and 231 between *Machilus yunnanensis* and *M. balansae* [44]) were identified in IR containing species, which were exceptionally smaller than the SNPs between two IR lacking species *T. alexandrinum* and *T. resupinatum* calculated in the present study. As an important structure in stabilizing cp genome, the IR region can prevent the genome mutating by selective force [45]. Thus the observed abundant SNPs/Indels in *T. alexandrinum* and *T. resupinatum* are not surprising.

The comparison between the Ka and Ks of genes is an important measure of molecular evolution [46]. Most genes were subjected to neutral selection and purification selection; however, there are also limited genes whose rate of Ka is higher than that of Ks because the function of the gene has been dramatically changed, called Darwinian positive selection [47]. Lacking one IR region is believed to directly enhance the nucleotide substitution rate of the single repeat sequence. Previous studies have shown that in the IR lacking cp genome, the nucleotide substitution rate in the remaining repeat region is comparable to that of the single repeat region, which is 2.3 times higher than that in the IR containing cp genome [48]. Here, seven protein-coding genes in the cp genomes of *T. alexandrinum* and *T. resupinatum* (*rps4*, *rpoC2*, *ndhG*, *ccsA*, *ndhF*, *ycf1* and *psaC*) have a high ratio of Ka to Ks, which is led by high values of Ka but extremely low values of Ks, and could imply that they are under positive selections. *rps4* [49] and *rpoC2* [50] have been reported to be under positive selection in previous studies. However, beneficial mutations might be fixed in those genes and, thus, reduce genetic polymorphism at selected sites [51].

In general, there is a strong correlation between the presence of IR and structurally stabilization of cp genomes. Substantial rearrangement was usually found in cp genomes lacking IR [13]. Among those IR lacking species of Leguminosae such as alfalfa, subclover, pea, and so forth, some are structurally stable and have not been rearranged, some undergo intermediate rearrangements, while others experienced a series of complex rearrangements [13]. This study found abundant rearrangements within fifteen cp genomes of IR lacking species of *Trifolium* (Figure 6). According to Palmer and Thompson [8], IR could prevent the rearrangement of cp sequence to some extent, so the rearrangement probability will be increased in the species lacking IR, this could be why there are many rearrangement events detected among those fifteen *Trifolium* species [8]. However, lacking IR leading to increased rearrangement is only one of the explanations. The repeats, acting as a locus of recombination within homologous genes, along with transposable elements (TEs), have been suggested as a reasonable mechanism for highly-rearranged cp genomes of *Trifolium* species [8].

### 3.5. Phylogeny Analysis and Divergence Time

The topological structure of other thirteen *Trifolium* species using 41 protein coding genes in this study was generally in agreement with the report of Sveinsson and Cronk [13] by 58 protein coding genes. In addition, the phylogenetic location of tested *T. alexandrinum* and *T. resupinatum* was confirmed (Figure 7). Furthermore, *T. alexandrinum* and *T. pratense*, both belonging to *Trifolium* Sect. Trifolium, were clustered together though *T. alexandrinum* and *T. subterraneum* were grouped together in Malaviya’s study based on isozyme data [52]. Three IR lacking species *T. boissieri*, *T. grandiflorum* and *T. aureum* were predicted to differentiate with other twelve species in the late Cretaceous period, then another two IR lacking species *T. strictum* and *T. glanduliferum* were further diverted at about 8 Mya. In late Cretaceous period, violent crustal movement and sea-land changes led to a flourished development of angiosperms and IR lacking species might form at the same time. It looks as if the ancestor of some IR lacking species had gone through a battery of evolutionary alternation (including high rearrangement and repetition) and the precise mechanism of such an evolutionary pattern is underway to illuminate.

## 4. Methods

### 4.1. Plant Material, DNA Isolation and Sequencing

Plant seeds of *T. alexandrinum* (cv ‘Elite II’) and *T. resupinatum* (cv ‘Laser’) were kindly provided by Barenbrug (Queensland, Australia) then germinated in a growth chamber (25 °C, 300 μmols·m^2^·s^−1^; 16-h photoperiod). Total DNA was extracted from 50 mg of fresh leaves following the Plant DNA Isolation Kit (Tiangen, Beijing, China). Sheared low molecular weight DNA fragments were used to construct paired-end (PE) libraries according to protocol of Illumina manual (San Diego, CA, USA). Completed libraries were pooled and sequenced in the Illumina NovaSeq platform with PE150 sequencing strategy and 350 bp insert size.

### 4.2. cp Genome Assembly, Annotation and Visualization

The raw read data for two *Trifolium* species were filtered according to the following criteria—reads of less than 5% unidentified nucleotides and more than 50% of their bases with a quality score of >20 were retained. With the reference genome of *T. meduseum* [12] (National Center for Biotechnology Information, NCBI number KJ 788288), the cp DNA were assembled as follows. In order to decrease the difficulty of sequences assembly, filtered reads (clean data) were aligned to the cp genome database built by Genepioneer Biotechnologies (Nanjing, China) using Bowtie2 v 2.2.4 [53] and SPAdes v3.10.1 [54] to acquire SEED sequences then obtained contigs by kmer iterative extend seed. Contigs, whose E-values were less than 1×10^−5^, Identities values were close to 100% and Gaps were close to 0 by a BLAST research in NCBI with the data set including *T. meduseum* [12], *T. pratense* [13], *T. repens* [12] and *T. subterraneum* [20], were obtained. Then the contigs were connected as scaffolds using SSPACE v 2.0 [55] followed by gap filling using Gapfiller v 2.1.1 [56] until the complete chloroplast genome sequence was recovered.

The results of CDS and rRNA were obtained using BLAST V 2.2.25 and HMMER V3.1 b2 and the cp genome database of NCBI. ARAGORN V 1.2.38 [57] and tRNAscan-SE search server (http://lowelab.ucsc.edu/tRNAscan-SE/, [58]) were used to predict and further check tRNA. Finally, the consensus annotation was obtained via Geneious (https://www.geneious.com, [59]) and visualized in OGDRAW (https://chlorobox.mpimp-golm.mpg.de/OGDraw.html, [60]).

### 4.3. The Relative Synonymous Codon Usage Analysis (RSCU) and Simple Sequence Repeats (SSRs) Prediction

The RSCU was analyzed using MEGA v7.0 to reflect the relative preference of a particular base encoding the corresponding amino acid codon [61]. Values of RSCU over one were considered to be a greater codon frequency. SSRs with the same repeats units and times and distributed in the genic regions were considered as shared repeats, the repetitive sequences were distinguished using VMATCH V2.3.0 (http://www.vmatch.de/) and MISA v1.0 (http://pgrc.ipk-gatersleben.de/misa/misa.html) based on the genomic data, which was also utilized to determine the mono-, di-, tri-, tetra-, penta- and hexa- nucleotides.

### 4.4. Sequence Variation Analysis and Ka/Ks

Whole cp genome alignment and collinearity analysis of sequenced species herein along with further thirteen IR-lacking *Trifolium* species, namely *T. grandiflorum* (NC_024034, [13]), *T. hybridum* (KJ788286, [13]), *T. lupinaster* (KJ788287, [13]), *T. occidentale* (KJ788289, [13]), *T. semipilosum* (KJ788291, [13]), *T. strictum* (NC025745.1, [13]), *T. aureum* (KC894708.1, [12]), *T. boissieri* (NC025743.1, [13]), *T. glanduliferum* (NC025744.1, [13]), *T. subterraneum* (NC011828, [20]), *T. meduseum* (NC476730.1, [13]), *T. pratense* (KJ788290, [13]) and *T. repens* (KC894706.1, [12]) was implemented using Mauve [62]. Furthermore, the common genes and shared protein-coding genes of *T. alexandrium* and *T. resupinatum* tested in the present study were utilized for nucleotide diversity (Pi) and Ka/Ks calculation. Ka/Ks, which was generally considered to be a reflection of selection pressures, was computed via KaKs_Calculator v2.0 [63]. Pi, which could be used to estimate the degree of nucleotide sequences variation and further provide potential molecular markers for population genetics, was calculated using VCFTOOLS [64] after sequences alignment of the common genes by MAFFT version 7.017 [65]. Finally, single nucleotide polymorphisms (SNPs) and insertions/deletions (In/Dels) of *T. alexandrium* and *T. resupinatum* were also identified using Mafft program [65].

### 4.5. Divergence Time Estimates

The 41 common genes sequence of fifteen *Trifolium* species and another ten Leguminosae species, including *Lotus japonicus* (AP002983.1), *Glycine max* (NC_007942.1), *Cicer arietinum* (NC_011163.1), *Ceratonia siliqua* (NC_026678.1), *Albizia odoratissima* (NC_034987.1), *Medicago truncatula* (KF241982.1), *M. sativa* (KU321683.1), *M. papillosa* (NC_027154.1), *M. hybrida* (NC_027153.1) and *Vicia sativa* (NC_027155.1) were first blasted using MEGA [61] then the alignment file was utilized to assess the divergence time using BEAST v 1.7.3 package [66] with the Bayesian method. GTR + G + I substitution model with a strict clock model and Yule model for Priors tree were applied for BEAUti along with MCMC analysis setting as follows, 10,000,000 of Chain length, 1000 of Tracelog, 1000 of screenlog, 1000 of treelog.t: tree. The assessment of results was executed in Tracer v 1.5 (http://www.beast.bio.ed.ac.uk/) to confirm that the value of effective sample size (ESS) was greater than 200. Finally, the tree file obtained from TreeAnnotator was visualized in Figtree v1.4.3 (http://tree.bio.ed.ac.uk/software/figtree/). Furthermore, phylogenetic trees of the fifteen *Trifolium* species based on *ycf1* gene and *rpoC2* gene were constructed under Maximum Composite Likelihood method with 1000 bootstrap replications using MEGA v7.0 [61].

## 5. Conclusions

cp genomes of *T. alexandrinum* and *T. resupinatum*, which belong to inverted-repeat-lacking clade (IRLC), were sequenced and annotated in present study and were compared with the cp genomes of other thirteen IR lacking *Trifolium* species reported previously. The results revealed abundant SNP and In/Del in *T. alexandrinum* and *T. resupinatum* cp genomes and high variation in CDS and abundant rearrangement within *Trifolium* genus. This valuable information will provide insight into the evolution of IR lacking species.

## Figures and Tables

**Figure 1 plants-09-00478-f001:**
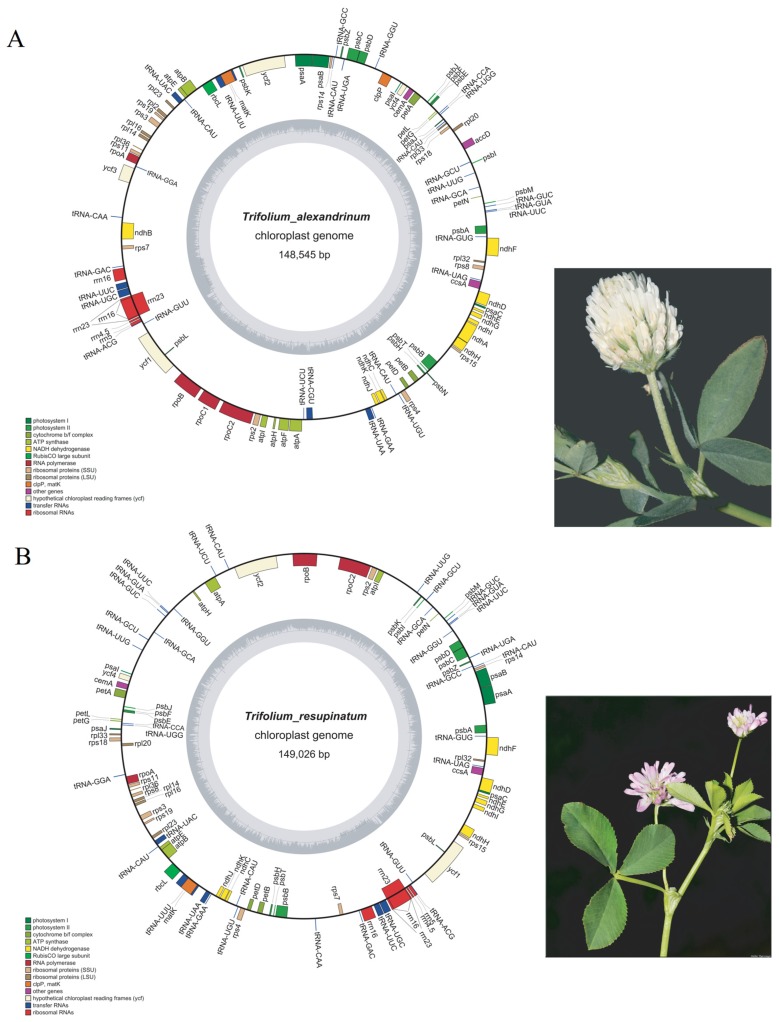
Gene maps of *T. alexandrinum* (**A**), *T. resupinatum* (**B**) cp genomes and pictures of *T. alexandrinum* and *T. resupinatum* were shown right in the gene maps. Pictures were cited from the open website http://plants-of-styria.uni-graz.at. Genes drawn inside and outside of the circle are transcribed clockwise and counterclockwise, respectively. Genes belonging to different functional groups are color coded. The darker gray color and lighter gray color in the inner circle corresponds to the GC content and the AT content, respectively.

**Figure 2 plants-09-00478-f002:**
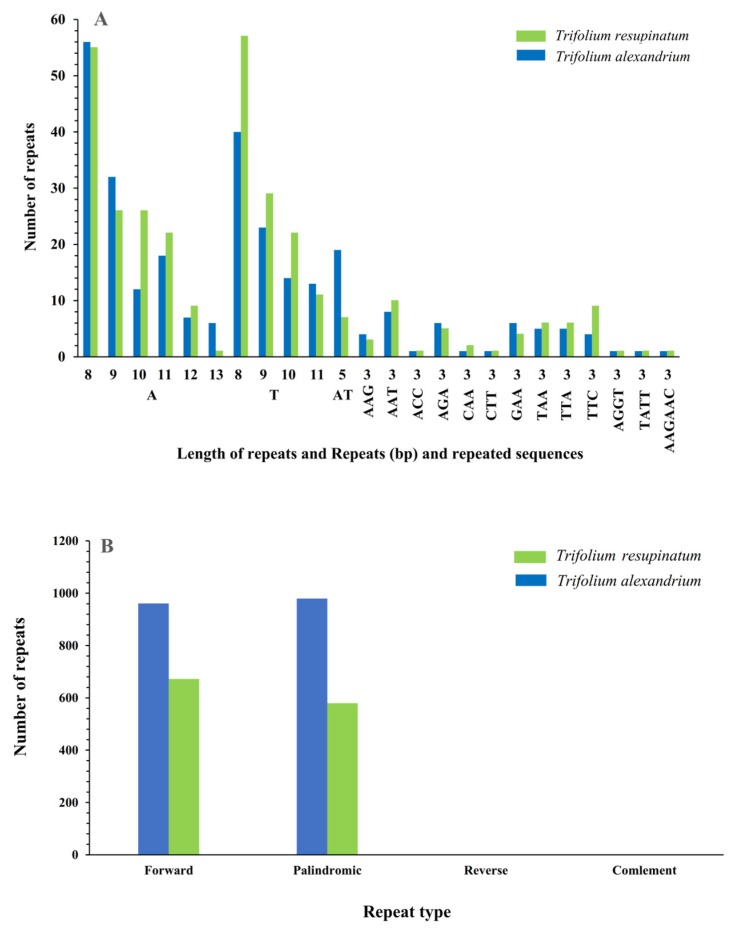
Repeating events of shared genes of *T. alexandrinum* and *T. resupinatum.* (**A**) Shared length of repeats and repeated sequences of *T. alexandrinum* and *T. resupinatum*; (**B**) Repeat type predicted in *T. alexandrinum* and *T. resupinatum* and (**C**) Listing of shared repetitive sequences with more than 30 bp.

**Figure 3 plants-09-00478-f003:**
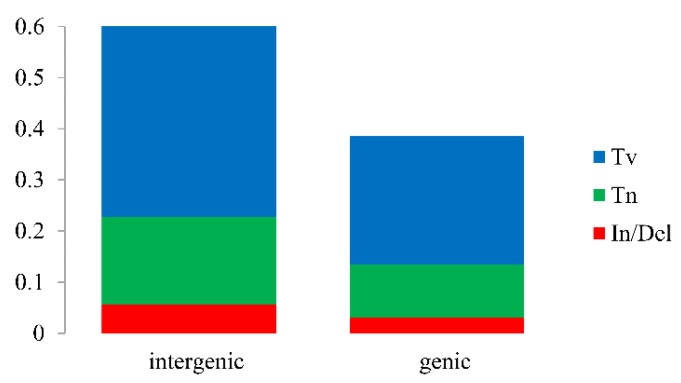
Transversion (Tv), transition (Tn) and Insert/Deletion (In/Del) were showed in intergenic and genic regions.

**Figure 4 plants-09-00478-f004:**
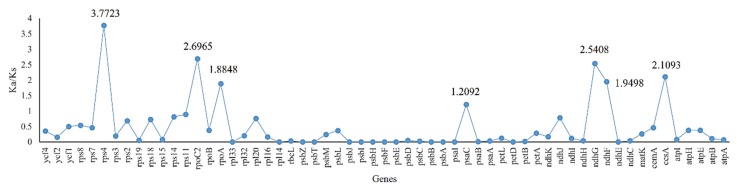
The synonymous/synonymous substitution rates (Ka/Ks) calculated using 59 shared genes in *T. alexandrinum* and *T. resupinatum*.

**Figure 5 plants-09-00478-f005:**
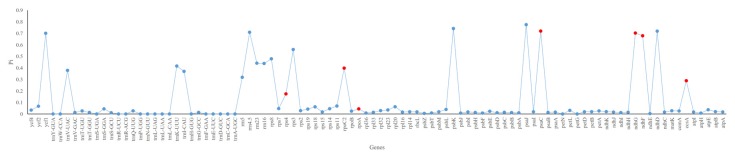
The nucleotide diversity (Pi) computed using 96 common genes of *T. alexandrinum* and *T. resupinatum.* Genes with Ka/Ks values more than one were red coded.

**Figure 6 plants-09-00478-f006:**
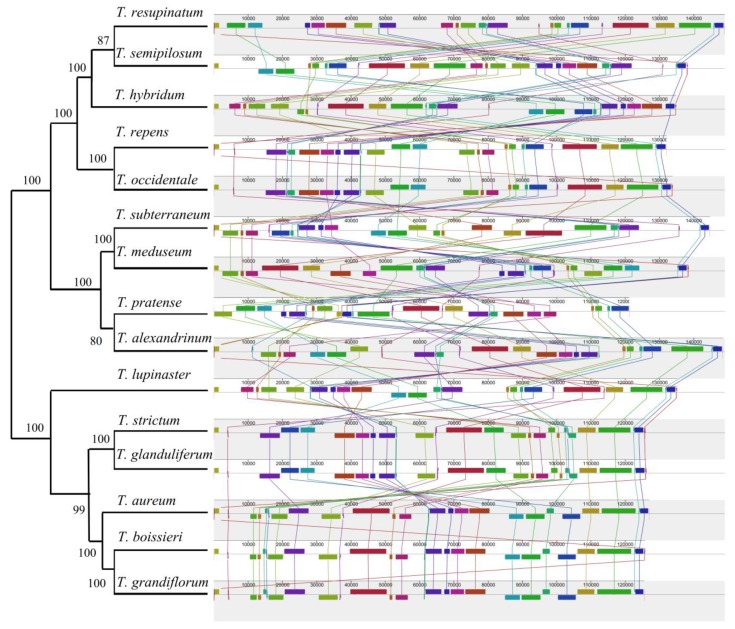
Synteny comparison of fifteen *Trifolium* chloroplast genomes with the reference of *T. resupinatum* using Mauve. Rectangular blocks with the same color indicate collinear regions.

**Figure 7 plants-09-00478-f007:**
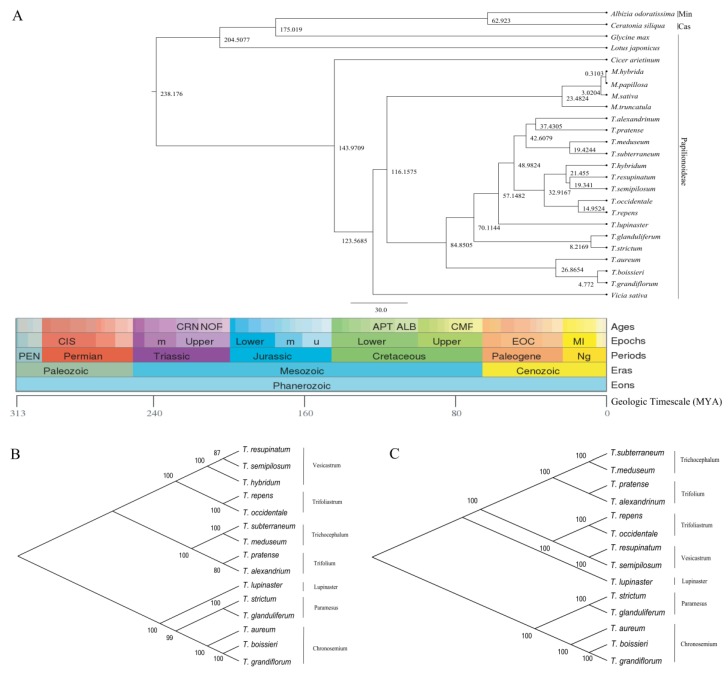
(**A**) BEAST chronogram of the 25 Leguminosae species based on the 41 common protein-coding genes and the phylogenetic trees constructed using *rpoC2* gene (**B**) and *ycf1* gene (**C**, *ycf1* gene was absent in *T. hybridum*). Geologic timescale was obtained from TIMETREE, time is shown in millions of years (MYA). Min, Mimosaceae; Cas, Caesalpinioideae. The over 50% bootstrap support values were showed in each main branch.

**Table 1 plants-09-00478-t001:** Comparison of the fifteen *Trifolium* species.

Species	Genome Length(bp)	GC Content (%)		Gene Density	tRNA	rRNA	mRNA	Genes	Genes with Introns	GenBank Number
	Repetitive %	cp Genome	CDS							
*T. alexandrinum*	148545/2.85%	34.09	37.05	7.54 × 10^−4^	31	6	75	112	13	MN857160
*T. resupinatum*	149026/2.69%	33.80	36.64	7.31 × 10^−4^	37	6	66	109	5	MN857161
*T. subterraneum*	144763/20.71%	34.83	37.10	7.60 × 10^−4^	30	4	76	110	16	NC011828
*T. meduseum*	142595/12.83%	34.87	37.34	7.78 × 10^−4^	30	4	77	111	15	NC476730.1
*T. pratense*	121178/NA *	34.63	36.94	7.43 × 10^−4^	28	4	58	90	11	KJ788290
*T. repens*	132120/20.70%	34.53	36.96	8.10 × 10^−4^	31	4	72	107	16	KC894706.1
*T.strictum*	125834/0.71%	34.98	36.70	8.82 × 10^−4^	31	5	75	111	18	NC025745.1
*T.aureum*	126970/5.60%	34.86	36.81	8.51 × 10^−4^	30	4	74	108	15	KC894708.1
*T.boissieri*	125740/1.05%	35.24	36.83	8.75 × 10^−4^	31	5	74	110	17	NC025743.1
*T.glanduliferum*	126149/0.78%	34.90	36.70	8.72 × 10^−4^	30	5	75	110	17	NC025744.1
*T. grandiflorum*	126149/5.60%	37.20	35.82	8.80 × 10^−4^	30	4	77	111	16	NC_024034
*T. hybridum*	134831/7.86%	34.33	35.97	8.08 × 10^−4^	31	4	74	109	17	KJ788286
*T. lupinaster*	135049/5.98%	33.97	35.71	8.15 × 10^−4^	30	4	76	110	15	KJ788287
*T. occidentale*	133780/4.64%	36.34	35.91	8.00 × 10^−4^	29	4	74	107	17	KJ788289
*T. semipilosum*	138194/10.55%	36.31	35.81	7.89 × 10^−4^	31	4	74	109	17	KJ788291

* The cp genome annotation of *Trifolium pratense* is incomplete, so the percentage of repetitive cannot be calculated [13].

**Table 2 plants-09-00478-t002:** List of genes annotated in the cp genomes of *T. alexandrinum* and *T. resupinatum.*

Category	Function	Name of Genes					
Self-replication (31)	Ribosomal RNA Genes	*rrn4.5*	*rrn5*	*rrn16*	*rrn23*		
	Transfer RNA genes	*trnA-ACG*	*trnA-GUC*	*trnA-GUU*	*trnA-UCU*	*trnA-UGC* ^*^	*trnC-GCA*
		*trnG-GCC*	*trnG-UUC* ^*^	*trnG-UUG*	*trnH-GUG*	*trnL-CAA*	*trnL-UAA* ^*^
		*trnL-UAG*	*trnL-UUU* ^*^	*trnM-CAU*	*trnP-GAA*	*trnP-UGG*	*trnS-GCU*
		*trnS-GGA*	*trnS-UGA*	*trnT-CCA*	*trnT-CGU*^*^ (ale)	*trnT-GGU*	*trnT-GUA*
		*trnT-UGU*	*trnV-GAC*	*trnV-UAC* ^*^			
Ribosomal proteins (10)	Small subunit of ribosome (SSU)	*rps2*	*rps3*	*rps4*	*rps7*	*rps8*	*rps11*
		*rps14*	*rps15*	*rps18* ^*/ale^	*rps19*		
Transcription (12)	Large subunit of ribosome (LSU)	*rpl2* (ale)	*rpl14*	*rpl16*	*rpl20*	*rpl23*	*rpl32*
		*rpl33*	*rpl36*				
	RNA polymerase subunits	*rpoA*	*rpoB*	*rpoC1*^*^ (ale)	*rpoC2*		
Photosynthesis related genes (46)	Large subunit of Rubisco	*rbcL*					
	Subunits of Photosystem I	*psaA*	*psaB*	*psaC*	*psaI*	*psaJ*	
	Subunits of Photosystem II	*psbA*	*psbB*	*psbC*	*psbD*	*psbE*	*psbF*
		*psbH*	*psbI*	*psbJ*	*psbK*	*psbL*	*psbM*
		*psbN* (ale)	*psbT*	*psbZ*			
	Subunits of ATP synthase	*atpA*	*atpB*	*atpE*	*atpF*^*^ (ale)	*atpH*	*atpI*
	Cytochrome b/f complex	*petA*	*petB*	*petD*	*petG*	*petL*	*petN*
	C-type cytochrome synthesis gene	*ccsA*					
	Subunits of NADH dehydrogenase	*ndhA*^*^ (ale)	*ndhB*^*^ (ale)	*ndhC*	*ndhD*	*ndhE*	*ndhF*
		*ndhG*	*ndhH*	*ndhI*	*ndhJ*	*ndhK*	
Other genes (7)	Maturase	*matK*					
	Protease	*clpP*^*^ (ale)					
	Chloroplast envelope membrane protein	*cemA*					
	Subunit of acetyl-CoA	*accD* (ale)					
	Hypothetical open reading frames	*ycf1*	*ycf2*	*ycf3*^**^ (ale)			

Note: ^*^, Genes containing a single intron; ^**^, Genes containing two introns; (ale), Genes that are particular for *T. alexandrinum*; ^*/ale^, Genes that only have an intron in *T. alexandrinum*.

## Data Availability

The annotated chloroplast genomes of *T. alexandrinum* and *T. resupinatum* have been deposited in the NCBI GenBank with the accession numbers MN857160 and MN857161.

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
