# Peer review of "The Complete Chloroplast Genome of Two Important Annual Clover Species, Trifolium alexandrinum and T. resupinatum: Genome Structure, Comparative Analyses and Phylogenetic Relationships with Relatives in Leguminosae"

_plants, 2020, doi:10.3390/plants9040478_

Round 1

Reviewer 1 Report

Overview: This study reported complete chloroplast (cp) genomes of two Trifolium species: T. alexandrinum and T. resupinatum. The authors presented the features of cp genomes of these species, performed comparative and phylogenetic analyses with other published Trifolium species. English writing is good, and results were well supported by tables, figures, and supplemental materials. In general, this manuscript is quite exciting, and I think this manuscript is an excellent addition to the plant genomics resource. I would like to see the paper published in the ‘plants’. I have a few minor suggestions regarding formatting:

Point 1-L37: Please provide both family name-Leguminosae (Fabaceae).

Point 2-P4 Table 1: I think that the mean values are not meaningful. How about deleting the mean values?

Point 3-P7 Figure 2A: Please change the format like Figure 2B., not the cumulative histogram.

Point 4-P9 Figure 5: The synonymous/synonymous substitution rates (Ka/Ks), which are above one within genus, are quite peculiar. Could you please consider these matters, including alpha error. Moreover, rps4 was calculated to 3.7723. Please discuss why protein-coding genes, rps4 calculated high values in detail, and what does this mean?

Point 5-P12 Figure 9: The phylogenetic tree from figure 9 should change to the higher resolution. Moreover, studied species grouped two clades in figure 9B and C. Are these clades related to the taxonomic group? (e.g., Infrageneric classification, subgeneric or sectional levels).

Point 6-P12 L111~112: If possible, please mention the correlation with gene transfer.

Point 7-P13 L129~131: I would like to compare and discuss for other Trifolium species, which is already published data.

Point 8-If possible, please provide plant photos of annual clover species, Trifolium alexandrinum, and T. resupinatum. It will be a great understanding these materials.

Author Response

All the modified parts have been marked in the manuscript with red. The responses to the reviewer's questions are as follows.

Question 1 L37: Please provide both family name-Leguminosae (Fabaceae)

Answer: Thanks for the reviewer's question. We have modified “Papilionoideae” into “Leguminosae, Fabaceae” in line 37.

Question 2 P4 Table 1: I think that the mean values are not meaningful. How about deleting the mean values?

Answer: Thanks for the reviewer's suggestion. We have deleted the mean values in Table 1.

Question 3 P7 Figure 2A: Please change the format like Figure 2B., not the cumulative histogram.

Answer: Thanks for the reviewer's suggestion. We have modified Figure 2A into the same format like Figure 2B.

Question 4 P9 Figure 5: The synonymous/synonymous substitution rates (Ka/Ks), which are above one within genus, are quite peculiar. Could you please consider these matters, including alpha error. Moreover, rps4 was calculated to 3.7723. Please discuss why protein-coding genes, rps4 calculated high values in detail, and what does this mean?

Answer: Thanks for the reviewer's question. The Ka/Ks values of most genes is less than 1, while Ka/Ks of some other genes (like rps4 gene) are significantly greater than 1, and the P value based on F test is not significant. This may be determined by the characteristics of Trifolium and that is what this study wants to reflect: there are abundant rearrangement (Fig 7) and variation (Fig 8) among Trifolium species. According to the research of Navarro et al [1], Ka/Ks value of the gene located in the rearrangement region is usually greater than 1. Such a gene may be rapidly evolving in the near future, which is important for the evolutionary study of Trifolium species and deserves to be further studied. Furthermore, we also calculated the Ka/Ks values of these genes again based on different software (like DNAsp) and algorithms, all of which showed a large Ka/Ks ratio (such as 2.9481 for rps4 gene and 1.7861 for pasC gene).

Reference: [1] Navarro A, Barton NH. Chromosomal speciation and molecular divergence-accelerated evolution in rearranged chromosomes. Science, 2003, 300(5617): 321-324.

Question 5: P12 Figure 9: The phylogenetic tree from figure 9 should change to the higher resolution. Moreover, studied species grouped two clades in figure 9B and C. Are these clades related to the taxonomic group? (e.g., Infrageneric classification, subgeneric or sectional levels).

Answer: Thanks for the reviewer's question. The phylogenetic tree from figure 9 was changed with higher resolution. Indeed, as described in “2.6. Phylogenetic Divergence Time Estimation” section, studied species grouped two clades in figure 9B and 9C. Section Lupinaster, Trifolium, Tricocephalum, Vesicastrum and Trifoliastrum were grouped into one clade in Fig 9B, which we call the “refractory clade”. And Sect. Paramesus and Subg. Chronosemium were grouped into another clade. In Fig 9C, however, Section Trifolium, Tricocephalum, Vesicastrum and Trifoliastrum were grouped into one clade, and another clade including Section Lupinaster, Paramesus and Subg. Chronosemium. This may be caused by the limitation of constructing phylogenetic tree only using single gene. Even so, each Section of Trifolium was clearly identified.

Question 6: P12 L111~112: If possible, please mention the correlation with gene transfer.

Answer: Thanks for the reviewer's question. We used gene transfer to further explain the differential genes between the two Trifolium species in line 118-121. The two cp genomes sequenced in this study revealed 75 and 66 protein-coding genes in T. alexandrinum and T. resupinatum, respectively (Table 1). There were nine genes (trnT-CGU, rpl2, rpoC1, atpF, ndhA, ndhB, ycf3, clpP and accD) unique in T. alexandrinum cp genome but absent in T. resupinatum, which might have been transferred to the nucleus genome of T. resupinatum. Similar result was revealed in tobacco with a absent of accD gene, which was essential for leaf development.

Question 7: P13 L129~131: I would like to compare and discuss for other Trifolium species, which is already published data.

Answer: Thanks for the reviewer's question. Indeed, we strongly recommend comparison and discussion with other already published data of other Trifolium species. For ndhB gene, there is no internal stop codon was detected in all the 14 (except T. resupinatum) Trifolium cp genomes, which are all the Trifolium species with the sequenced cp genome. However, ndhB gene contains an internal stop codon in other legume species. Thus we compare and discuss with legumes species, other than Trifolium species in line 139-141.

Question 8: If possible, please provide plant photos of annual clover species, Trifolium alexandrinum, and T. resupinatum. It will be a great understanding these materials.

Answer: Thanks for the reviewer's suggestion. The pictures of Trifolium alexandrinum and T. resupinatum were provided in Fig 1 and showed with (C) and (D).

Reviewer 2 Report

The manuscript of Xiong et al. reports two chloroplast genomes from Trifolium species. The authors compared the cp genomes with other existing Trifolium species and provide divergence time of the genus. Despite numerous analyses have been conducted, the patterns of genome rearrangements in Trifolium remain unclear and the findings are lack of novelty. This article would be of high interests if the authors could provide more insights into mechanisms of genome structural variation.

Comments-

  1. As the Trifolium chloroplast genomes are rich in repeats and the genome structure are dramatically rearranged. It is concerned whether the genomes were assembled and annotated accurately or not. Please provide a graph showing each genome assembly and the read depth of each site to ensure each assembly had sufficient coverage.
  2. The methodology of genome sequencing is not clear, please provide details of the sequencing strategy such as the sequencing platform and the read length.
  3. Fig. 1 – The annotation of rrn16 and rrn23 is questionable. Please annotate each gene carefully.
  4. Some of the published Trifolium cp genomes (e.g. in Sveinsson and Cronk, 2014) are not included in analysis. Please include all the sequences if possible.
  5. Some of the results are lack of discussion and interpretation such as codon usage analysis (Fig. 3).
  6. Fig. 7 – Please provide the gene synteny blocks with the gene names. It seems these genomes are highly rearranged, but it is difficult for readers to track with genes and knowing the patterns of rearrangement (if present).
  7. Ln191 “explain is a verb”. And since the authors suggest IR lacking is only one of the explanations to gene rearrangement, it would be helpful if the authors can provide other explanations in detail at the same time.
  8. Table 1 – the repetitive % of pratense is not available. Please provide the numbers if this is a public data.

Author Response

All the modified parts have been marked in the manuscript with red. The responses to the reviewer's questions are as follows.

Question 1: As the Trifolium chloroplast genomes are rich in repeats and the genome structure are dramatically rearranged. It is concerned whether the genomes were assembled and annotated accurately or not. Please provide a graph showing each genome assembly and the read depth of each site to ensure each assembly had sufficient coverage.

Answer: Thanks for the reviewer's question. Indeed, the accurate assembly and annotation are important for this manuscript. The graphs of genomic coverage are added in the supplementary data (Fig S1).

Question 2: The methodology of genome sequencing is not clear, please provide details of the sequencing strategy such as the sequencing platform and the read length.

Answer: Thanks for the reviewer's question. The detail of genomic sequencing was further described and the sequencing platform and the read length were also added in “4.1. Plant Material, DNA Isolation and Sequencing” section.

Question 3: Fig. 1 – The annotation of rrn16 and rrn23 is questionable. Please annotate each gene carefully.

Answer: Thanks for the reviewer's question. Indeed, because the sequences of rrn16 and rrn23 genes had overlaps, so they were not well annotated in the previous uploaded Figure 1. Therefore, we reannotated rrn16 and rrn23 in Fig 1.

Question 4: Some of the published Trifolium cp genomes (e.g. in Sveinsson and Cronk, 2014) are not included in analysis. Please include all the sequences if possible.

Answer: Thanks for the reviewer's question. All the 15 sequenced Trifolium species were added in our analysis (Table 1, Fig 7, Fig 8, and Fig 9), including two species sequenced here (Trifolium alexandrinum and T. resupinatum) and 13 species reported before (T. aureum, T. boissieri, T. glanduliferum, T. grandiflorum, T. hybridum, T. lupinaster, T. meduseum, T. occidentale, T. pratense, T. repens, T. semipilosum, T. strictum and T. subterraneum).

Question 5: Some of the results are lack of discussion and interpretation such as codon usage analysis (Fig. 3).

Answer: Thanks for the reviewer's question. The discussion about codon usage analysis was added in “3.2 Relative synonymous codon usage analysis (RSCU)” section.

Question 6: Fig. 7 – Please provide the gene synteny blocks with the gene names. It seems these genomes are highly rearranged, but it is difficult for readers to track with genes and knowing the patterns of rearrangement (if present).

Answer: Thanks for the reviewer's suggestion. Indeed, if we can show the gene name in the Fig 7, it will facilitate the reader's reading. However, we are so sorry for we can’t show the gene name, because different colored blocks mean locally colinear blocks (LCB), they may contain several genes. What we wanted to show by synteny analysis that there was a lot of rearrangement among Trifolium species, and of course the genes associated with rearrangement are what we're going to be studied in the future. Moreover, this kind of figures were made in other similar published papers about cp genome, such as references [1] and [2].

Reference:

[1] Kaila, T.; Chaduvla, P.K.; Rawal, H.C.; Saxena, S.; Tyagi, A.; Mithra, S.V.; Gaikwad, K. Chloroplast genome sequence of clusterbean (Cyamopsis tetragonoloba L.): genome structure and comparative analysis. Genes. 2017, 8(9), 212.

[2] Du Q, Bi G, Mao Y, et al. The complete chloroplast genome of Gracilariopsis lemaneiformis, (Rhodophyta) gives new insight into the evolution of family Gracilariaceae. Journal of Phycology, 2016, 52(3):441-450.

Question 7: Ln191 “explain is a verb”. And since the authors suggest IR lacking is only one of the explanations to gene rearrangement, it would be helpful if the authors can provide other explanations in detail at the same time.

Answer: Thanks for the reviewer's question. We have changed “explains” into “explanations” in line 210. A lot of rearrangement was found among Trifolium species in this study, one of the explanations was the IR-lacking. Furthermore, as described in “3.4. Sequence Divergence and Hotspots” section, the repeats, acting as a locus of recombination within homologous genes, along with transposable elements (TEs), have been suggested as a reasonable mechanism for highly-rearranged cp genomes of Trifolium species.

Question 8: Table 1 – the repetitive % of pratense is not available. Please provide the numbers if this is a public data.

Answer: Thanks for the reviewer's question. According to Sveinsson S and Cronk Q, the Cp genome structure of T. pratense appears to be highly complex and they were unable to complete a full assembly of it, thus they did not predict the repeat ratio of the T. pratense Cp genome in their paper [1]. Furthermore, since the Cp genome sequence of T. pratense has not been fully assembled, we consider it meaningless to calculate the repetition ratio of T. pratense.

Reference:

[1] Sveinsson, S.; Cronk, Q. Evolutionary origin of highly repetitive plastid genomes within the clover genus (Trifolium). BMC Evol. Biol. 2014, 14(1), 218-228.

Round 2

Reviewer 2 Report

The revised manuscript of Xiong et al. have followed the reviewer’s suggestion to include more Trifolium for comparison, however, there are some concerns regarding to the results and the analytical methods.

  1. Please provide another coverage graph. Despite the authors have provided a graph showing genomic coverage, (Fig. S1), the figure is unclear due to lack of a figure legend and clear labels. And if the coverage plot is the green circle shown, it seems some of the areas are with none or very low read mapping. If so, then additional PCR verifications are required. I would recommend the authors provide a coverage graph that is shown in linear form for better visualization. And we can ensure each site have received sufficient coverage.
  2. Fig. 1 – The annotation of rrn16 and rrn23 are controversial. The authors suggested there are two rrn16 genes in each of the cp genomes and one of the rrn16 is overlapped and reverse complement to its rrn23. This is a special and rare case if the finding is true. It is unclear why this only happens in the two cp genomes reported but not in other Trifolium. The authors should discuss this novel feature in the text.
  3. The plant images shown in the revised Fig. 1 C-D are beautiful, but the size can be smaller. Please make sure all text in the figures to be at least 5 points in size. The text shown on Fig. 1A-B are too small to read.
  4. Fig. 7 – The phylogeny is important for tracking back evolutionary history. It would be helpful if the authors could provide a simplified phylogeny (from the Fig. 9) left to the Fig.7 and sort the synteny order according to its phylogenetic position. It is easier for readers to track with the rearrangement events.
  5. Fig. 8 – Given the cp genomes are highly rearranged (Fig. 7), I have no idea how the genome alignments were performed in mVISTA analysis? The authors need to provide the details of the alignment strategy. It is unclear why the authors used T. resupinatum as reference and how come the missing genes (rpl2, ycf3 and so on) are shown in T. resupinatum? The gene labels also seem questionable to me as rrn16 and rrn23 are labeled as exon.
  6. Fig. 9 – The phylogeny of Fig. 9A is difficult to visualize. Please enlarge the phylogram and the text on it. What sequences dataset are included in the phylogenetic analysis? For all the sequences or just the genic regions? Again, the authors need to provide the details in methods.
  7. Method - Because several genes are unable to identify in T. resupinatum and many of these missing genes are essential for survival. Please provide the details of the BLAST criteria. What is the e-values for gene search and what are the genomes included in database? In addition, what cause further loss of genes in T. resupinatum? Is there any explanation?
  8. It is helpful to include more Trifolium species in analysis, but some of the numbers in abstract, introduction and methods need to be updated as well.
  9. The reported results lack logical cohesion. All the results should connect to specific research questions. Therefore, if the results (such as the codon usages) are important but not connect to main questions, please show as supplementary data.

Author Response

All the modified parts have been marked with blue in the manuscript. The responses to the reviewer's questions are as follows.

Question:(1)Please provide another coverage graph. Despite the authors have provided a graph showing genomic coverage, (Fig. S1), the figure is unclear due to lack of a figure legend and clear labels. And if the coverage plot is the green circle shown, it seems some of the areas are with none or very low read mapping. If so, then additional PCR verifications are required. I would recommend the authors provide a coverage graph that is shown in linear form for better visualization. And we can ensure each site have received sufficient coverage.

Answer: Thanks for the reviewer’s suggestion. We have provided linear coverage graphs in Fig S2 to reflect the sufficient coverage in each site. Indeed, it’s difficult to assemble cp genomes of T. alexandrinum and T. resupinatum because the cp genomes of those two species contain many repeat fragment and abundant rearrangement among the related Trifolium species, leading the relatively low coverage in some site. However, we succeed in assembling them ultimately referred to cp genome of T. meduseum, a clover species with relatively large number of genes and the relatively long cp genome, and we also found that most of the low coverage area is located in the gene spacer, and genic region generally contains higher coverage.

Question:(2)Fig. 1 – The annotation of rrn16 and rrn23 are controversial. The authors suggested there are two rrn16 genes in each of the cp genomes and one of the rrn16 is overlapped and reverse complement to its rrn23. This is a special and rare case if the finding is true. It is unclear why this only happens in the two cp genomes reported but not in other Trifolium. The authors should discuss this novel feature in the text.

Answer: Thanks for the reviewer’s question. We reannotated the chloroplast genomes of two Trifolium species and also found that there are two rrn16 in each of Trifolium species sequenced in this study, and this phenomenon was also reported in cp genomes of T. strictum (NC025745.1) and T. glanduliferum (NC025744.1). Furthermore, we also found that two rrn23 weren’t reverse complement (81979 bp to 84755 bp and 84773 bp to 81978 bp in T. alexandrium, 122934 bp to 125730 bp and 125712 bp to 122935 bp in T. resupinatum), and rrn16 and rrn23 were still partially overlapped. Indeed, no similar phenomenon was found in other Trifolium species. Previous studies have shown that Cyanobacteria, Chlamydomonas reinhardtii, Cyanophora paradoxa, Zea mays, Oryza sativa and Arabidopsis thaliana can start and stop transcription anywhere in the chloroplast genome [1]. Chloroplasts are known to originate from ancient Cyanobacteria, this transcription property may be conserved in some species like T. alexandrinum and T. resupinatum, so that the rRNA genes of T. alexandrinum and T. resupinatum may gain the potential transcription ability. On the other hand, the chloroplast genomes of most plant species are small and in order to avoid costs, the genes overlapping is happened. In the same way, this content was also added in Discussion section from line 248-258.

Reference:

[1] Shi C, Wang S, Xia E H, et al. Full transcription of the chloroplast genome in photosynthetic eukaryotes. Scientific Reports, 2016, 6: 30135.

Question:(3)The plant images shown in the revised Fig. 1 C-D are beautiful, but the size can be smaller. Please make sure all text in the figures to be at least 5 points in size. The text shown on Fig. 1A-B are too small to read.

Answer: Thanks for the reviewer’s question. We have changed the plant images in Fig1 with smaller size and enlarged the text on Fig1 A-B.

Question:(4)Fig. 7 – The phylogeny is important for tracking back evolutionary history. It would be helpful if the authors could provide a simplified phylogeny (from the Fig. 9) left to the Fig.7 and sort the synteny order according to its phylogenetic position. It is easier for readers to track with the rearrangement events.

Answer: Thanks for the reviewer’s suggestion. We have provided a phylogeny left to the Fig 6 (Synteny comparison) and sorted the synteny order according to the phylogenetic position.

Question:(5)Fig. 8 – Given the cp genomes are highly rearranged (Fig. 7), I have no idea how the genome alignments were performed in mVISTA analysis? The authors need to provide the details of the alignment strategy. It is unclear why the authors used T. resupinatum as reference and how come the missing genes (rpl2, ycf3 and so on) are shown in T. resupinatum? The gene labels also seem questionable to me as rrn16 and rrn23 are labeled as exon.

Answer: Thanks for the reviewer’s question. The mVISTA analysis was applied to sequence alignment among several related species of angiosperms [1] to reflect the degree of genic variation. According to the annotation files (GFF3) of each cp genomes, the genic information was extracted from their own cp genome sequences (Fasta), and genic variation was finally detected follow the LAGAN alignment program. However, we re-verified the previous mVISTA results and found that there was indeed too much incorrectness of these Trifolium species, such as positional fault of unique genes and base position mismatching. Selecting any one Trifolium species as a reference does not fully annotate the genes of other species. Therefore, we chose T. resupinatum, a species with relatively large number of genes and the longest chloroplast genome, as the reference genome. In addition, we set the mVISTA analysis order of those 15 Trifolium species according to the phylogenetic position shown in Fig. 7A, so that the reader can track the evolution of each gene. Furthermore, we are so sorry rrn16 and rrn23 were manually incorrect annotated as exon referring to Liu et al. [1] in the annotation files (GFF3) when carrying out the mVISTA analysis. Because the Trifolium species are highly rearranged, we considered that the mVISTA analysis was not suitable for the Trifolium species and recommended to delete it. So, mVISTA relevant result has been deleted in the manuscript.

Reference:

[1] Liu, L.X.; Wang, Y.W., He, P.Z.; Li, P.; Lee, J.; Soltis, D.E.; Fu, C.X. Chloroplast genome analyses and genomic resource development for epilithic sister genera Oresitrophe and Mukdenia (Saxifragaceae), using genome skimming data. BMC Genomics, 2018, 19(1):235.

Question:(6)Fig. 9 – The phylogeny of Fig. 9A is difficult to visualize. Please enlarge the phylogram and the text on it. What sequences dataset are included in the phylogenetic analysis? For all the sequences or just the genic regions? Again, the authors need to provide the details in methods.

Answer: Thanks for the reviewer’s suggestion. The size of Fig 9A and the text have been enlarged and the picture has been renamed Fig 7A. As described in “4.5. Divergence Time Estimates” section from line 407-412, the 41 common genes sequence of 15 Trifolium species and another 10 Leguminosae species, including Lotus japonicus (AP002983.1), Glycine max (NC_007942.1), Cicer arietinum (NC_011163.1), Ceratonia siliqua (NC_026678.1), Albizia odoratissima (NC_034987.1), Medicago truncatula (KF241982.1), M. sativa (KU321683.1), M. papillosa (NC_027154.1), M. hybrida (NC_027153.1) and Vicia sativa (NC_027155.1) were firstly blasted using MEGA then the alignment file was utilized to assess the divergence time using BEAST v 1.7.3 package.

Question:(7)Method - Because several genes are unable to identify in T. resupinatum and many of these missing genes are essential for survival. Please provide the details of the BLAST criteria. What is the e-values for gene search and what are the genomes included in database? In addition, what cause further loss of genes in T. resupinatum? Is there any explanation?

Answer: Thanks for the reviewer’s question. As described in “4.2. Cp Genome Assembly, Annotation and Visualization” section (from line 372-374), a BLAST research in NCBI with the data set including cp genomes of T. meduseum, T. pratense, T. repens and T. subterraneum, was conducted according to the E-values were less than 1e-5, Identities values were close to 100% and Gaps were close to 0. As described in “3.1. Genome Feature of T. alexandrinum and T. resupinatum and Comparison with Other IR Lacking Trifolium Species” (from line 236-240), there were nine protein-coding genes unique in T. alexandrinum cp genome but absent in T. resupinatum, one of the reasons for gene loss in T. resupinatum was gene transfer, and gene transfer of accD [1] and rpl2 [2] was also occurred in other species. Another reason for gene loss (like ndhA and ndhB) was nutritional status and rearrangement [3]. Gene loss of other four genes (atpF [4], psbN [4], rpoC1 [4] and ycf3 [5]) absent in T. resupinatum was also found in other species. However, the detailed reason for those gene loss was remained to be explained.

Reference:

[1] Rousseau-Gueutin, M.; Huang, X.; Higginson, E., Ayliffe, M.; Day, A.; Timmis, J.N. Potential functional replacement of the plastidic acetyl-CoA carboxylase subunit (accD) gene by recent transfers to the nucleus in some angiosperm lineages. Plant physiology. 2013, 161(4), 1918-1929.

[2] Adams Keith L, Ong Han Chuan, Palmer Jeffrey D. Mitochondrial Gene Transfer in Pieces: Fission of the Ribosomal Protein Gene rpl2 and Partial or Complete Gene Transfer to the Nucleus. Molecular Biology & Evolution, 2001(12), 12.

[3] Kim, H. T.; Kim, J. S.; Moore, M. J.; Neubig, K. M.; Williams, N. H.; Whitten, W. M.; Kim, J.H. Seven new complete plastome sequences reveal rampant independent loss of the ndh gene family across orchids and associated instability of the inverted repeat/small single-copy region boundaries. PloS One. 2015, 10, e0142215.

[4] Xi, L.; Zhang, T.C.; Qiao, Q.; Ren, Z.M.; Zhao, J.; Yonezawa, T.; Hasegawa, Masami.; C Crabbe, M.J.; Li, J.Q.; Zhong, Y. Complete chloroplast genome sequence of holoparasite Cistanche deserticola (Orobanchaceae) reveals gene loss and horizontal gene transfer from its host Haloxylon ammodendron (Chenopodiaceae). Plos One, 2013, 8(3): e58747-.

[5] Cai, Z.Q.; Guisinger, M.; Kim, H.G.; Ruck, E.; Blazier, J.C.; McMurtry, V.; Kuehl, J.V.; Boore, J.; Jansen, R.K. Extensive reorganization of the plastid genome of Trifolium subterraneum (Fabaceae) is associated with numerous repeated sequences and novel DNA insertions. Journal of Molecular Evolution. 2008, 67(6), 696-704.

Question:(8)It is helpful to include more Trifolium species in analysis, but some of the numbers in abstract, introduction and methods need to be updated as well.

Answer: Thanks for the reviewer’s question. We have updated the numbers in the line 25, 26 169, 268 and 293.

Question:(9)The reported results lack logical cohesion. All the results should connect to specific research questions. Therefore, if the results (such as the codon usages) are important but not connect to main questions, please show as supplementary data.

Answer: Thanks for the reviewer’s suggestion. Indeed, the RSCU result may not connect to main questions in this study, so we have changed it into Fig S1 in Additional file1.

Round 3

Reviewer 2 Report

The manuscript has improved considerably, and here are some minor comments.

  1. 1 – The authors confirmed there are two rrn16 genes in the two reported Trifolium species and also T. strictum and T. glanduliferum, but it is unclear if the two rrn16 sequences are identical in each of the species? Please specify the findings in the results.

  1. The gene labels in the Fig. 1 are still blurry. Please enlarge the words shown.

  1. 6 – The genome synteny is too small to read. Please narrow the width of the simplified phylogeny (see the example showed in reference provided).

     Hamsher SE, Keepers KG, Pogoda CS, Stepanek JG, Kane NC, Kociolek JP (2019) Extensive chloroplast genome rearrangement amongst three closely related Halamphora spp. (Bacillariophyceae), and evidence for rapid evolution as compared to land plants. PLoS ONE 14(7): e0217824. https://doi.org/10.1371/journal.pone.0217824

  1. Page2, Ln47 – please rephrase “What’s more”.

Author Response

All the modified parts have been marked with red in the manuscript. The responses to the reviewer's questions are as follows.

Question 1: The authors confirmed there are two rrn16 genes in the two reported Trifolium species and also T. strictum and T. glanduliferum, but it is unclear if the two rrn16 sequences are identical in each of the species? Please specify the findings in the results.

Answer: Thanks for the reviewer’s question. Indeed, there are two rrn16 genes in the two reported Trifolium species and also T. strictum and T. glanduliferum, but the length of two rrn16 genes was different in each of the two Trifolium species (1493bp and 2484bp in T. alexandrinum, 1491bp and 2485bp in T. resupinatum) sequenced in this study and also T. strictum (1490bp and 73bp) and T. glanduliferum (1490bp and 73bp). We also specify this finding in “2.1. Features of the T. alexandrium and T. resupinatum cp Genomes” section in line 98.

Question 2: The gene labels in the Fig. 1 are still blurry. Please enlarge the words shown.

Answer: Thanks for the reviewer’s question. All the words in Fig.1 were enlarged.

Question 3: The genome synteny is too small to read. Please narrow the width of the simplified phylogeny (see the example showed in reference provided). Hamsher SE, Keepers KG, Pogoda CS, Stepanek JG, Kane NC, Kociolek JP (2019) Extensive chloroplast genome rearrangement amongst three closely related Halamphora spp. (Bacillariophyceae), and evidence for rapid evolution as compared to land plants. PLoS ONE 14(7): e0217824. https://doi.org/10.1371/journal.pone.0217824

Answer: Thanks for the reviewer’s suggestion. The genome synteny (Fig 6) was enlarged and the width of the simplified phylogeny was narrowed according to Hamsher et al.

Question 4: Page2, Ln47 – please rephrase “What’s more”.

Answer: Thanks for the reviewer’s question. The sentence “What’s more, it is also very important as a park, garden and green place plant” was changed to “What’s more, it also contains economic significance to the ornamental and landscape industries” in line 47.
